# Effects of COVID-19 Infection during Pregnancy and Neonatal Prognosis: What Is the Evidence?

**DOI:** 10.3390/ijerph17114176

**Published:** 2020-06-11

**Authors:** Álvaro Francisco Lopes de Sousa, Herica Emilia Félix de Carvalho, Layze Braz de Oliveira, Guilherme Schneider, Emerson Lucas Silva Camargo, Evandro Watanabe, Denise de Andrade, Ana Fátima Carvalho Fernandes, Isabel Amélia Costa Mendes, Inês Fronteira

**Affiliations:** 1Global Health and Tropical Medicine, Instituto de Higiene e Medicina Tropical, Universidade Nova de Lisboa, 1349-008 Lisboa, Portugal; ifronteira@ihmt.unl.pt; 2Human Exposome and Infectious Diseases Network (HEID), Escola de Enfermagem de Ribeirão Preto, Universidade de São Paulo, 14040-902 Ribeirão Preto, Brazil; hericacarvalho@usp.br (H.E.F.d.C.); layzebraz@usp.br (L.B.d.O.); guilherme.schneider@usp.br (G.S.); lucmrg0@gmail.com (E.L.S.C.); ewatanabe@forp.usp.br (E.W.); dandrade@eerp.usp.br (D.d.A.); iamendes@usp.br (I.A.C.M.); 3Nursing Department, Universidade Federal do Ceará, 60430-160 Fortaleza, Brazil; afcana@ufc.br

**Keywords:** COVID-19, SARS-CoV-2, pregnancy, fetal transmission, mother-to-child transmission

## Abstract

Background: This study’s aims are to assess the current evidence presented in the literature regarding the potential risks of COVID-19 infection among pregnant women and consequent fetal transmission. Methods: a systematic literature review assessing papers published in the most comprehensive databases in the field of health intended to answer the question, “What are the effects of COVID-19 infection during pregnancy, and what is the neonatal prognosis?” Results: 49 papers published in 2020 were eligible, presenting low levels of evidence. A total of 755 pregnant women and 598 infants were assessed; more than half of pregnant women had C-sections (379/65%). Only 493 (82%) infants were tested for SARS-CoV-2, nine (2%) of whom tested positive. There is, however, no evidence of vertical transmission based on what has been assessed so far, considering there are knowledge gaps concerning the care provided during and after delivery, as well as a lack of suitable biological samples for testing SARS-CoV-2. Conclusions: We cannot rule out potential worsening of the clinical conditions of pregnant women infected with SARS-CoV-2, whether the infection is associated with comorbidities or not, due to the occurrence of respiratory disorders, cardiac rhythm disturbances, and acid-base imbalance, among others. We recommend relentless monitoring of all pregnant women in addition to testing them before delivery or the first contact with newborns.

## 1. Introduction

On 30 January 2020, the World Health Organization (WHO) declared the outbreak of COVID-19, a respiratory disease caused by the new coronavirus SARS-CoV-2, as the sixth public health emergency of international concern [1,2] Due to its highly transmissible nature, by 9 April 2020, it had spread to five continents, and approximately 85,522 people had died [2].

Considering that transmission seems to mainly occur through contact with respiratory droplets [3] produced by an infected person, anticipating public health measures intended to control and prevent the infection, such as adherence to universal precautions, quarantine, and timely diagnosis, are options available to mitigate the transmission of COVID-19 [4]. 

Clinical manifestations range from asymptomatic cases and mild upper airway infection, up to severe and fatal cases with pneumonia and acute respiratory failure [5,6,7]. This variation is because people with prior diseases/comorbidities are less apt to fight the virus so that it is more likely to reach the lungs and cause pneumonia. Elderly individuals with comorbidities such as noncommunicable diseases and immunocompromised persons are at the highest risk of developing signs and symptoms of COVID-19 and having them worsened [5,6].

It is, however, unknown how COVID-19 infection behaves in key populations more commonly susceptible to viral diseases, such as pregnant women [8], as well as whether there is the possibility of vertical transmission or premature birth. 

The changes in the immune system of pregnant women make them more susceptible to infectious processes, in addition to the manifestations of the infection, with the risk of adverse maternal and neonatal complications, premature birth, spontaneous abortion, application of endotracheal intubation, restriction of intrauterine growth, hospitalization in an intensive care unit, renal failure, intravascular coagulopathy, and transmission to the fetus or newborn [9].

Current studies on the susceptibility of pregnant women to infection by COVID-19 are still incipient and adopt poor methods, and although transmission of the virus to the fetus or baby during delivery or pregnancy has not been proven, the presence of antibodies has already been identified, namely, specific IgG for viruses in neonatal serum samples [10].

Due to the need to provide evidence for clinical practice involving pregnant women, this study’s objective is to assess current evidence presented in the literature regarding the potential risks of COVID-19 infection among pregnant women and consequent fetal transmission.

## 2. Materials and Methods 

This systematic literature review [11], with no protocol registration, is intended to answer the question, “What are the effects of COVID-19 infection during pregnancy and what is the neonatal prognosis?” The PECO [12] method was adopted, in which

Population (P) = pregnant women;Exposure (E) = COVID-19 infection;Comparison (C) = has not been an object of study;Outcome (O) = maternal and/or fetal infection by SARS-CoV-2.

A search was conducted in the following databases: US National Library of Medicine (PubMed), Scopus, Embase, ScienceDirect (Elsevier), Web of Science (WoS), Scholar Google, and preprints servers bioRxiv and medRxiv, as well as the bibliographic references of the selected papers (hand searching). These databases were selected due to their range and representativeness in the field of basic and health sciences. Terms that derived from the following expressions were used according to the databases/servers: “COVID-19” OR “SARS-CoV-2” AND “Pregnancy” AND “Perinatal”. To avoid screening biases, two researchers with expertise in the method and topic under study independently and concomitantly searched all the databases on 25 and 26 May. The researchers had a discussion to reach a consensus about which papers would be included or excluded from the study, and a third reviewer mediated disagreements that prevented them from reaching a consensus. 

Observational epidemiological studies and case reports addressing the clinical conditions of mother–fetus pairs and including primary data of patients over 18 years old were considered eligible. Manuscripts that contained only data from pregnant women, or only fetuses, or that did not address the period of delivery, such as puerperium, were disregarded. No restrictions regarding the period of publication or language were imposed. Review papers, opinions reports, local reports, abstracts of events, and similar works were excluded. Social, demographic, and clinical data included in the studies were collected.

The GRADE system was used to classify levels of evidence. The results are presented in terms of prevalence calculated in the study and combined results.

## 3. Results

In total, 148 studies were initially identified, 20 of which were excluded because they were repeated. Of the 128 remaining studies, 56 were excluded for not addressing this review’s objective. The full texts of 72 studies were analyzed, although 23 did not meet the eligibility criteria, so 49 studies remained and composed the final sample of this systematic review (Figure 1).

The studies were divided into 21 case reports (42%) [13,14,15,16,17,18,19,20,21,22,23,24,25,26,27,28,29,30,31,32,33], 19 cross-sectional descriptive studies (40%) [34,35,36,37,38,39,40,41,42,43,44,45,46,47,48,49,50,51,52], 7 cross-sectional analytical studies (14%) [53,54,55,56,57,58,59], 1 case–control study (2%) [60], and 1 cohort study (2%) [61] (Table 1). 

In Table 2, we present the characteristics of pregnant women and newborns, with social, demographic, and clinical data being collected.

### 3.1. Findings from Case Reports

The 21 case reports (42%) [13,14,15,16,17,18,19,20,21,22,23,24,25,26,27,28,29,30,31,32,33] addressed 29 women in the second (17%) and third trimester (83%) of pregnancy, admitted with signs and symptoms of COVID-19 (97%), later confirmed through RT-PCR (100%). Twenty-two (75%) developed mild to moderate pneumonia. The majority of the studies were conducted in China (50%) [13,14,15,17,20,21,22,23,25,28,31] and presented limited description of the main features of the disease in positive, pregnant women. As for comorbidities, more than half of the women did not present any comorbidity (63%) [13,17,19,21,22,23,24,25,27,28,29,31]. Among those with comorbidities, the most common was fetal distress (16%) [16,20,22,23,32]. The most frequent clinical signs and symptoms were fever (72%) [13,14,16,19,22,23,24,25,26,27,28,29,30,31,33] and cough (62%) [15,16,19,20,21,22,23,25,26,27,29,31,32,33]. Among the imaging findings, chest scans revealed pulmonary changes (96%) [13,14,15,16,17,19,20,21,22,23,25,26,27,29,30,31,32,33]. Laboratory exams revealed increased reactive C protein (69%) [14,15,17,19,20,23,25,27,29,30,31] and decreased lymphocytes (lymphocytopenia) (58%) [13,14,17,19,20,22,23,26,27,28,29,31]. Twenty-seven infants were born from twenty-seven women, predominantly through C-section delivery (78%) [13,14,15,16,17,20,21,22,23,25,26,28,29,30,33], there was medical indication to half of these due to maternal comorbidities [14,15,16,20,22,23,26,30] or infection [13,17,20,23,25,28,29]. Of the 27 newborns, two died [27,32], one of these died along with his/her mother, and 26 were tested for SARS-CoV-2: two tested positive [30,33]. In 13 cases (48%), newborns were isolated from mothers [13,14,16,17,19,20,21,28,30,33], and, in six cases, the placenta was analyzed for pathological alterations [13,17,32,33]. There were no confirmed cases of vertical transmission.

### 3.2. Findings from Descriptive Studies

In cross-sectional descriptive studies (40%) [34,35,36,37,38,39,40,41,42,43,44,45,46,47,48,49,50,51,52], 16 (85%) were conducted in China [34,35,36,37,38,39,40,41,42,44,46,47,49,50,51,52]. A total of 546 pregnant women were assessed for COVID-19, 409 (75%) of whom were diagnosed with the disease through RT-PCR. The majority (85%) was in the third trimester of pregnancy and had mild to moderate pneumonia (88%). Most women did not present any comorbidity (35%) [34,35,36,38,39,40,42,44,45,46,47,48,50,52]. The signs and symptoms more frequently found in pregnant women were fever [34,35,36,39,40,41,42,43,44,45,46,48,49,50,51,52] and cough [34,35,36,38,39,40,41,42,43,44,45,46,47,48,49,50,51,52]. Imaging findings revealed suggestive images of infection in 377 (93%) pregnant women [35,42,44,46,47,48,50,51,52] and laboratory exams showed increased reactive C-protein (56%) [35,36,38,39,40,41,42,44,45,46,47,48,49,50,51] and lymphocytopenia (40%) [35,36,38,39,40,41,42,44,45,46,47,48,50,51]. A total of 421 pregnancies resulted in 429 newborns, with 8 twin births. C-section was the most frequent type of delivery (64%), the majority resulting from pregnancy comorbidities (46%) [34,35,36,38,39,40,41,42,43,45,46,47,50]. Six newborns died after birth [34,48] and two were stillbirths [42,48]. Seven mothers died from severe respiratory complications [48]. Three hundred and forty-five newborns were tested for SARS-CoV-2, seven of whom tested positive (2%) [36,45,46,47]. In 97 [43,45,46,52] cases, the newborns were isolated from their mothers. A total of 32 placentas were analyzed, with no abnormal findings [46].

### 3.3. Findings from Cross-Sectional Analytical Studies

Concerning the cross-sectional analytical studies (14%) [53,54,55,56,57,58,59], six were conducted in China (86%) and one was conducted in the United States of America (14%). A total of 133 pregnant women were infected with COVID-19, confirmed through laboratory (78%) and clinical diagnosis (22%). Most pregnant women were in their third trimester (80%) and had developed mild to moderate pneumonia (99%). Clinical findings showed that only 41 (35%) [54,55,56,57,59] women had some comorbidity (Table 2). The most common signs and symptoms were fever at admission [55,56,57,58,59], postpartum fever [54,57,58], and cough [54,55,56,58]. 

Chest CT scan was suggestive in 96 (95%) women [54,55,56,57,58] and laboratory exams revealed increased reactive C protein (78%) [54,55,56], lymphocytopenia (77%) [54,55,56], and neutrophilia (77%) [54,56]. Of the 106 women, 59 (65%) had a C-section, 19 of whom due to COVID-19 [53,57]. A total of 108 babies were born, with two sets of twins. Eighty-seven (94%) were tested for SARS-CoV-2, and all were negative. Twenty-three pregnant women were isolated from their newborns [55,58]. There were no neonatal deaths; 16 placentas were analyzed, and no pathognomonic features were identified. Vertical transmission was not confirmed.

### 3.4. Findings from Longitudinal Studies

The case–control study (2%) [60] was conducted using the medical records of pregnant women admitted to a hospital in China. The study compares the clinical features, maternal and neonatal outcomes of 16 pregnant women with COVID-19, and 18 without the disease, but suspected of being infected. The study does not clearly report criteria for including participants in the case or control groups. The description in Table 2 refers to COVID-19 infected pregnant women. All 16 women were in their third trimester of pregnancy, 5 (31%) of whom had no pregnancy-related comorbidities. The most frequent symptom was fever at admission (25%) and after birth (50%). Ten women had suggestive CT scans, and there was an increase in reactive C protein and neutrophilia in all women studied. Fourteen (87%) women had a C-section, but the study does not report why a C-section was indicated. A total of 17 babies were born, with one set of twins, with no complications. Only three newborns were tested for SARS-CoV-2 and were negative. Isolation measures after birth or analysis of the placenta were not reported.

The cohort study (2%) [61], also conducted in China, retrospectively describes 31 pregnant women and 35 non-pregnant women with COVID-19. Only the clinical findings of pregnant women are described. In total, 31 pregnant women were assessed. The majority was in the third trimester of pregnancy (71%), all had confirmation of diagnosis through RT-PCR, 21 (68%) developed mild to moderate pneumonia, and 10 had severe pneumonia. Twenty-eight did not present comorbidities during pregnancy. The most prevalent signs and symptoms were fever (55%) and cough (48%). The chest CT scan was suggestive in all cases, and abnormal laboratory tests were related to the increased number of neutrophils (32%), aspartate transaminase (26%), and interleukin 6. There were 17 deliveries, 13 (76%) of which were C-sections, though the authors do not report why C-sections were indicated. A total of 17 single, healthy fetuses were born and tested negative for SARS-CoV-2. No report is provided on the isolation measures adopted between mothers and babies after delivery and there was no evaluation of placentas.

## 4. Discussion

This review was intended to answer a question concerning the effects of COVID-19 infection during pregnancy and neonatal prognosis. Forty-nine studies were eligible and included case reports, cross-sectional, analytical cross-section, case–control, and cohort presenting low levels of evidence. The low levels of evidence are due to the novelty of the COVID-19 pandemic and the need for rapidly acquiring knowledge to support public policies. As the number of cases increases worldwide, evidence about the impact of this virus during pregnancy for both women and newborns is expected to become stronger, especially with the development of more robust comparative studies and follow-up with control groups. 

### 4.1. Characterization of Pregnant Women

A total of 755 pregnant women were assessed, 635 were from China [13,14,15,17,20,21,22,23,25,28,31,34,35,36,37,38,39,40,41,42,44,46,47,49,50,51,52,53,54,55,56,57,58,60,61], 60 from the USA [19,42,59], 42 from Italy [45], 10 from Iran [27,49], and one pregnant woman in Asia [16], Honduras [18], Australia [24], Turkey [26], Spain [29], Peru [30], Switzerland [32], and Canada [33], respectively. All women evaluated were in the fertile period, while there were pregnant women in the first trimester (46/6%), in the second trimester (77/10%) and in the third trimester of pregnancy (632/84%), which is why only 598 infants were born in the period. 

The fact that most pregnant women were from China imposes limitations on interpreting the evidence, considering cultural and epidemiological differences when compared with pregnant women from other countries and cultures. However, even in the minority (120/16%), non-Chinese pregnant women were evaluated, and the characteristics (clinical and epidemiological) showed no differences.

Regarding the pregnant women’s age, fertile period, and length of pregnancy, the studies analyzed showed a wide variation and a lack of evidence of infection by SARS-CoV-2 during the first and second trimester of pregnancy. It can be inferred that, according to the low prevalence of severe infection among pregnant women (57/8%), many of them could be asymptomatic and/or with mild symptoms, without the need for hospital care, corroborating findings from previous viral pandemics [62].

In view of the limited data regarding the Middle East Respiratory Syndrome (MERS), a systematic review with meta-analysis recovered seven studies that did not report spontaneous abortion. The rate of premature birth was 32.1% (3 of 11), all occurring before 34 weeks of gestation. Preeclampsia was described in 19.1% (1 of 7); however, no cases of premature rupture of membranes or restricted fetal growth were reported. The rates of C-sections and perinatal death were 61.8% (5 of 8) and 33.2% (3 of 10), respectively, including two stillbirths and one neonatal death (4 h after the birth of an extremely premature baby). There were no reports of fetal distress, Apgar score <7 at 5 min, neonatal asphyxia, or admission to the neonatal intensive care unit (ICU). Finally, signs of vertical transmission were not found during the follow-up period in any of the newborns [63].

According to the above, it appears that the scarce data on infection with the new coronavirus in early pregnancy may be related, but not limited, to the absence of tests performed during this period, since asymptomatic cases may go unnoticed due to poorer surveillance of pregnancy because of restrictions, on the part of medical staff or the pregnant women themselves, to attend appointments within a hospital setting. Thus, we suggest that tests for COVID-19 should be routinely performed in prenatal care.

### 4.2. Clinical Findings in Pregnant Women

About diagnosis, this study verifies that of the 728 pregnant women evaluated, 589 (81%) and 139 (19%) were diagnosed by Reverse Transcription followed by Polymerase Chain Reaction (RT-PCR) and by clinical assessment, respectively. The gold standard for diagnosing Covid-19 is tissue culture in which the antigen is isolated, using Polymerase Chain Reaction (PCR), which detects nucleic acid. Even so, a single result not detected through RT-PCR for SARS-CoV-2 does not exclude a COVID-19 diagnosis, as there are various factors, such as inadequate sample collection, type of biological sample, the time elapsed between sample collection and onset of symptoms, and fluctuation of viral load, that may influence a test’s result. For this reason, an RT-PCR test should be repeated in another sample of a patient’s respiratory tract whenever there are discordances between results and epidemiological conditions, especially in populations where a false-positive may result in disastrous consequences.

The presence of comorbidities related to pregnancy does not seem to directly influence the adverse outcomes of pregnant women and their newborns, as the two neonatal deaths were of mothers without comorbidities, but who, for some reason, developed severe pneumonia. However, it is observed that gestational diabetes and fetal distress were the most prevalent comorbidities, showing that the conditions of the fetus should be carefully evaluated, especially in those asymptomatic and without comorbidities. Thus, the absence of comorbidities may directly influence the care provided and attention paid by professionals to pregnant women, exposing some of them to a worse prognosis [64].

Concerning the signs and symptoms of infection at the time of admission, 689 (91%) pregnant women were evaluated [13,14,15,16,17,18,19,20,21,22,23,24,25,26,27,28,29,30,31,32,33,34,35,36,37,38,39,40,41,42,43,44,45,46,47,48,49,50,51,52,53,54,55,56,57,58,59,60,61], and the main signs and symptoms presented were fever at admission (363/53%), cough (290/42%), and dyspnea (83/12%). It is remarkable that 83 (12%) of pregnant women were asymptomatic but were tested (through RT-PCR to detect SARS-CoV-2) due to exposure to people diagnosed with COVID-19, reinforcing the need to follow recommendations provided by the Centers for Disease Control and Prevention (CDCs) to testing risk groups in contact with those diagnosed with COVID-19, [65] though this approach may not be feasible in some contexts where there is a shortage of tests.

Regarding imaging tests, chest CT scans were performed in 577 (76%) pregnant women [13,14,15,16,17,18,19,20,21,22,23,24,25,26,27,28,29,30,31,32,33,34,35,36,37,38,39,40,41,42,43,44,45,46,47,48,49,50,51,52,53,54,55,56,57,58,59,60,61], with 538 (93%) pregnant women showing changes suggestive of infection [13,14,15,16,17,19,20,21,22,23,25,26,27,29,30,31,32,33,35,42,44,48,50,51,52,55,56,57,58,60,61]. CT scans were very useful in the initial assessment at the time of admission. The most prevalent changes were in the findings of C-reactive protein, which was above the normal range [14,15,17,19,20,23,25,27,29,30,31,35,36,38,39,40,41,42,44,45,48,49,50,51,60], and lymphocytopenia [13,14,17,19,20,22,23,26,27,28,29,31,35,36,38,39,40,41,42,44,45,48,50,51,60]. It is outstanding that these changes in CT scans and laboratory findings are reported in studies with the general population [66]. Concerning the evaluation of imaging tests, whether they are chest X-rays or chest CT scans, these can assist in the diagnosis of the disease; however, they should not be taken as conclusive evidence for confirming or excluding SARS-CoV-2 infection. The reason is that different bacterial and viral etiologic agents cause pulmonary infections. Thus, imaging tests such as chest CT scans, widely used during emerging respiratory outbreaks or as in the case of the pandemic currently established by COVID-19, have high sensitivity, but low specificity. Despite limitations, these tests should be used to screen, evaluate, and monitor this kind of infection [67].

### 4.3. Childbirth

Overall, 587 childbirths were reported, with more than half of the pregnant women having C-sections (379/65%). When checking the indications for the C-sections, pregnancy-related comorbidities were the cause for performing C-sections in 148 pregnant women; in the case of 103, the indication was infection, and in 128 cases, indications were not reported. No details were provided when the indication for a C-section was the infection. In most cases, the time of delivery was determined by obstetric reasons, rather than the maternal diagnosis of COVID-19 [68].

The decision for the type of delivery is usually more frequently influenced by the presence of maternal and/or fetal impairment. When there were imminent risks, an emergency C-section was the alternative chosen, which has happened in the case of SARS-CoV-2 infections in which the pregnant woman’s clinical condition is complex. However, in the presence of COVID-19, the threshold for C-sections became lower than usual so that infection control procedures could be more easily adhered to and the transmission of the disease to the fetus is minimized [69].

### 4.4. Newborns

Regarding the characteristics of the newborns, 598 babies were born, with 590 single births and eight twins. A total of 493 (82%) newborns were tested for SARS-CoV-2, nine (2%) of whom tested positive, 101 (20%) were premature, and 28 (6%) were born underweight (<2500 g). Ten neonatal deaths [27,32,34,42,48] and one spontaneous abortion [42] were reported. Isolation measures were established between 130 mothers and their newborns, while no pathognomonic changes (indicative of disease) were found in the 54 placental samples assessed.

Note that most of the newborns did not present serious complications. The unfavorable outcomes refer to ten neonatal deaths, one spontaneous abortion, eight maternal deaths, one stillbirth, nine (2%) positive SARS-CoV-2 tests, and three newborns with high rates of IgG and IgM antibodies against SARS-CoV-2.

Regarding the nine newborns who tested positive for SARS-CoV-2, three tested positive [33,48] immediately after delivery, but then tested negative 24 h after delivery. Another newborn tested positive 36 h after delivery [36]. It is important to note that whether the mothers and babies in these three cases were isolated after delivery is not reported, making it difficult to establish the transmission routes, according to the authors, in addition to the absence of intrauterine tissue and amniotic fluid samples.

Two newborns tested positive for the new coronavirus after being breastfed by their mothers without wearing masks, as the maternal infection was not known in the postpartum period. However, vertical transmission could not be confirmed or ruled out, as the authors stressed that the newborns were not tested for COVID-19 immediately after birth. Moreover, in this same study, a pregnant woman with COVID-19 gave birth to a newborn by vaginal delivery, who tested positive for SARS-CoV-2 despite the mother wearing a surgical mask and the medical team wearing proper PPE throughout labor [45]. In another study [70], a possible transmission through breast milk was raised by the detection of viral RNA in the milk of one of the pregnant women and subsequent confirmation of contamination of the baby, although the mother followed safety precautions when breastfeeding the child. This reinforces the need for testing mothers before giving birth, using proper vestments (staff and mother), and avoiding breastfeeding when testing is not possible.

The other two newborns tested positive 16 h [30] and 53 h [46] after delivery, respectively. Both studies reported the followed guidelines were observed: the use of masks by the mother, the entire medical team was attired, mother and baby were isolated after delivery, and no breastfeeding. The authors, however, reported an important limitation of the study, the absence of complementary assessments (presence of viruses in amniotic fluid, umbilical cord blood, or placental tissue). Two out of the nine positive babies were intubated [30,45], and three had mild pneumonia [36,46,47], although they fully recovered within a few days.

Dong et al. [28] reported the case of a newborn who, shortly after birth, presented leukocytosis and a high rate of IgG and IgM antibodies against SARS-CoV-2 and IL-6 cytokines. The baby, however, presented no symptoms and tested negative for the virus. Although the newborn tested negative for SARS-CoV-2, the authors stated that the high rate of IgM antibodies within 2 h after birth suggested the occurrence of intrauterine infection, as there would be no transfer of these antibodies from the mother to the fetus through the placenta due to the size of this macromolecule. Additionally, in general, these take three to seven days to be produced by the body after contact with the infectious agent. The authors, however, do not rule out the possibility of infection during delivery.

Zeng et al. [35] described similar results regarding the assessment of the presence of specific antibodies against SARS-CoV-2 in the blood of newborns from mothers with confirmed COVID-19 infection. Two babies presented rates of IgG and IgM antibodies specific to the virus above the normal level, but none showed symptoms of the infection. The authors of this study stressed the possibility that the newborn developed IgM antibodies during the gestational period if the virus had crossed the placental barrier.

### 4.5. Limitations

The studies recovered in this review have several limitations. Among them, small sample sizes, retrospective evaluation of medical records with incomplete data, as well as the low number of babies tested for SARS-COV-2, compared to the number of births.

Among this review’s limitations, we highlight the possibility that the same patients are reported more than once, as they may have been included in different studies and we addressed secondary data only. Most of the cases reported refer to Chinese pregnant women and there was a lack of a standard assessment in newborns to verify the presence of viruses in amniotic fluid, umbilical cord blood or placental tissue. There was also little information about the care provided during delivery and postpartum, and an absence of clear inclusion criteria for the control group. Nonetheless, despite these limitations, this review has strengths, such as presenting a comparison of all studies available so far, compiling scattered data and grouping them more clearly in order to make analyses and inferences, and, based on what was assessed, reporting that there is no evidence of vertical transmission thus far, as there are knowledge gaps concerning the care provided during and after delivery, and a lack of suitable biological samples for testing SARS-CoV-2. 

## 5. Conclusions

The results show that the potential worsening of the women infected with SARS-CoV-2 cannot be ruled out, whether the infection is associated with comorbidities or not. As documented here, there is a risk of women developing respiratory disorders or having cardiac rhythm disturbances or acid-base imbalance, among others. The risk of postpartum hemorrhage and premature delivery is significant, which is why C-sections were widely used. Relentless monitoring is recommended for pregnant women who report signs and symptoms suggested of COVID-19, especially at a time when women in countries with radical isolation measures have difficulty attending regular prenatal care. Those who cannot be tested before delivery should avoid having contact with the baby and wear masks to decrease the spread of the disease.

Regarding the risk for babies, there is no concrete evidence of vertical transmission, though one cannot discard this possibility. Nonetheless, cases of respiratory diseases, abnormal Apgar indexes, and mild pneumonia were reported. Fortunately, all the babies who tested positive for the infection recovered fully and rapidly. A rigorous assessment of the newborns’ clinical signs is recommended, as well as chest CT scans, within three days after birth.

## Figures and Tables

**Figure 1 ijerph-17-04176-f001:**
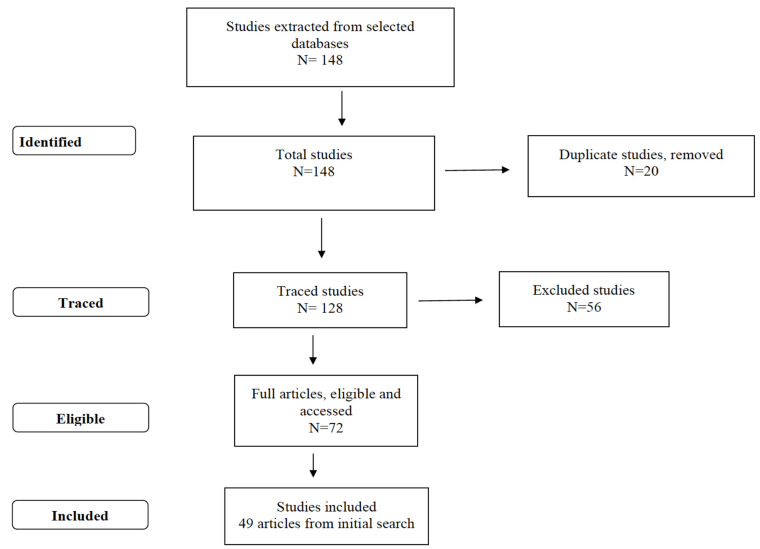
Study selection flowchart.

**Table 1 ijerph-17-04176-t001:** Characteristics of included studies: reference, study design, location, level of evidence, and limitations.

Code	Reference	Study Design	Location	Level of Evidence	Limitations
1	Fan et al. (2020) [13]	Case report	China	Very Low	Small sample size; single setting; only third-trimester pregnant women; without long-term follow-up
2	Chen et al. (2020) [14]	Case report	China	Very Low	Small sample size; single setting; only third-trimester pregnant women; without long-term follow-up
3	Li et al. (2020) [15]	Case report	China	Very Low	Small sample size; single setting; only third-trimester pregnant women; without long-term follow-up
4	Lee et al. (2020) [16]	Case report	Asia	Very Low	Small sample size; single setting; only third-trimester pregnant women; without long-term follow-up
5	Xiaotong et al. (2020) [17]	Case report	China	Very Low	Small sample size; single setting; only third-trimester pregnant women; without long-term follow-up
6	Zambrano et al. (2020) [18]	Case report	Honduras	Very Low	Small sample size; single setting; only third-trimester pregnant women; without long-term follow-up
7	Iqbal et al. (2020) [19]	Case report	United States of America	Very Low	Small sample size; single setting; without long-term follow-up; without additional assessments of the virus in amniotic fluid, umbilical cord blood or placenta tissue
8	Kang et al. (2020) [20]	Case report	China	Very Low	Small sample size; single setting; only third-trimester pregnant women; without long-term follow-up
9	Lu et al. (2020) [21]	Case report	China	Very Low	Small sample size; single setting; only third-trimester pregnant women; without long-term follow-up
10	Liao et al. (2020) [22]	Case report	China	Very Low	Small sample size; only third-trimester pregnant women; without information on the delivery or the isolation conditions of newborns after delivery
11	Buonsenso et al. (2020) [23]	Case report	China	Very Low	Small sample size; single setting; only third-trimester pregnant women; without long-term follow-up
12	Lowe e Bopp (2020) [24]	Case report	Australia	Very Low	Small sample size; single setting; only third-trimester pregnant women; without long-term follow-up
13	Khan et al. (2020) [25]	Case report	China	Very Low	Small sample size; single setting; only third-trimester pregnant women; without long-term follow-up
14	Kalafat et al. (2020) [26]	Case report	Turkey	Very Low	Small sample size; single setting; only third-trimester pregnant women; without long-term follow-up
15	Karami et al. (2020) [27]	Case report	Iran	Very Low	Small sample size; single setting; only third-trimester pregnant women; without long-term follow-up
16	Dong et al. (2020) [28]	Case report	China	Very Low	Small sample size; single setting; only third-trimester pregnant women; without long-term follow-up
17	González et al. (2020) [29]	Case report	Spain	Very Low	Small sample size; only third-trimester pregnant women; without information on the delivery or the isolation conditions of the newborn after delivery
18	Alzamora et al. (2020) [30]	Case report	Peru	Very Low	Small sample size; single setting; without long-term follow-up; without additional assessments of the virus in amniotic fluid, umbilical cord blood or placenta tissue
19	Xiong et al. (2020) [31]	Case report	China	Very Low	Small sample size; single setting; only third-trimester pregnant women; without long-term follow-up
20	Baud et al. (2020) [32]	Case report	Switzerland	Very Low	Small sample size; single setting; only third-trimester pregnant women; without long-term follow-up
21	Kirtsman et al. (2020) [33]	Case report	Canada	Very Low	Small sample size; single setting; only third-trimester pregnant women; without long-term follow-up
22	Liu et al. (2020) [34]	Cross-sectional descriptive	China	Very Low	Small sample size; single setting; only third-trimester pregnant women; without long-term follow-up
23	Chen et al. (2020) [35]	Cross-sectional descriptive	China	Very Low	Small sample size; single setting; only third-trimester pregnant women; without long-term follow-up
24	Yu et al. (2020) [36]	Cross-sectional descriptive	China	Very Low	Small sample size; single setting; only third-trimester pregnant women; without long-term follow-up
25	Zeng et al. (2020) [37]	Cross-sectional descriptive	China	Very Low	Small sample size; single setting; only third-trimester pregnant women; without long-term follow-up
26	Chen, Liao and Shao (2020) [38]	Cross-sectional descriptive	China	Very Low	Small sample size; single setting; only third-trimester pregnant women; without long-term follow-up
27	Liu et al. (2020) [39]	Cross-sectional descriptive	China	Very Low	Small sample size; single setting; only third-trimester pregnant women; without long-term follow-up
28	Zhu et al. (2020) [40]	Cross-sectional descriptive	China	Very Low	Small sample size; single setting; only third-trimester pregnant women; with little information on the exams performed
29	Chen et al. (2020) [41]	Cross-sectional descriptive	China	Very Low	Small sample size; without long-term follow-up
30	Yan et al. (2020) [42]	Cross-sectional descriptive	China	Very Low	Small sample size; single setting; without long-term follow-up
31	Breslin et al. (2020) [43]	Cross-sectional descriptive	United States of America	Very Low	Small sample size; without long-term follow-up
32	Chen et al. (2020) [44]	Cross-sectional descriptive	China	Very Low	Small sample size; single setting; without long-term follow-up
33	Ferrazzi et al. (2020) [45]	Cross-sectional descriptive	Italia	Low	Without long-term follow-up; not all newborns were tested immediately after birth; without additional assessments of the virus in amniotic fluid, umbilical cord blood or placenta tissue
34	Nie et al. (2020) [46]	Cross-sectional descriptive	China	Very Low	Small sample size; not all newborns were tested for SARS-CoV-2; without long-term follow-up.
35	Khan et al. (2020) [47]	Cross-sectional descriptive	China	Very Low	Small sample size; single setting; only third-trimester pregnant women; without long-term follow-up.
36	Hantoushzadeh et al. (2020) [48]	Cross-sectional descriptive	Iran	Very Low	Small sample size; single setting; only third-trimester pregnant women; without long-term follow-up.
37	Qianchenga et al. (2020) [49]	Cross-sectional descriptive	China	Very Low	Small sample size; single setting; only third-trimester pregnant women; without long-term follow-up; absence of criteria for allocation to groups
38	Wu et al. (2020) [50]	Cross-sectional descriptive	China	Very Low	Small sample size; single setting; without long-term follow-up
39	Yang et al. (2020) [51]	Cross-sectional descriptive	China	Very Low	Small sample size; single setting; without additional assessments of the virus in amniotic fluid, umbilical cord blood or placenta tissue
40	Liu et al. (2020) [52]	Cross-sectional descriptive	China	Very Low	Small sample size; single setting; only third-trimester pregnant women; without long-term follow-up; absence of criteria for allocation to groups
41	Zhang et al. (2020) [53]	Cross-sectional analytical	China	Low	Small sample size; single setting; lack of clear criteria to include control groups; little information on postpartum care provided to newborns
42	Liu et al. (2020) [54]	Cross-sectional analytical	China	Low	Small sample size; no clear criteria for inclusion in the control group; absence of tomography images for monitoring therapeutic effects in ambulatory pregnant women
43	Liao et al. (2020) [55]	Cross-sectional analytical	China	Low	Small sample size; single setting; without long-term follow-up; only third-trimester pregnant women
44	Wu et al. (2020) [56]	Cross-sectional analytical	China	Very Low	Small sample size; single setting; without long-term follow-up
45	Yue et al. (2020) [57]	Cross-sectional analytical	China	Low	Small sample size; single setting; only third-trimester pregnant women
46	Yang et al. (2020) [58]	Cross-sectional analytical	China	Low	Small sample size; single setting; only third-trimester pregnant women; not all newborns were tested for SARS-CoV-2
47	Shanes et al. (2020) [59]	Cross-sectional analytical	United States of America	Low	Small sample size; single setting; not all newborns were tested for SARS-CoV-2
48	Lin et al. (2020) [60]	Case-control	China	Very Low	Small sample size; single setting; only third-trimester pregnant women; retrospective; without clear inclusion criteria for cases and controls
49	Yin et al. (2020) [61]	Cohort	China	Low	Small sample size; single setting; without long-term follow-up; incomplete information on pregnant women; controls are 35 non-pregnant women of fertile age

**Table 2 ijerph-17-04176-t002:** Characteristics of included studies: study design, settings and participants, clinical, laboratory, and imaging findings of women, and type of delivery and clinical and laboratory findings of newborns.

Code	Study Design	Setting and Participants	Clinical, Laboratory and Imaging Findings of Women and Type of Delivery	Newborns’ Clinical and Laboratory Findings
[13,14,15,16,17,18,19,20,21,22,23,24,25,26,27,28,29,30,31,32,33]	Case report	Setting: China (11/50%), Asia (1/5%), The Netherlands (1/5%), United States of America (1/5%), Turkey (1/5%), Iran (1/5%), Peru (1/5%), Italy (1/5%), Australia (1/5%), Switzerland (1/5%), and Canada (1/5%); - Pregnant women: 29 COVID-19 infected pregnant women with laboratory confirmation (29/100%), in the second (5/17%) or third (24/83%) trimester of pregnancy, mild or moderate (22/75%) and severe (5/17%) pneumonia; - 27 newborns: tested for SARS-CoV-2 (26/96%).	- Pregnancy associated comorbidity (27/93% assessed): no comorbidity (17/63%), fetal distress (5/16%), placenta previa (2/7%), gestational diabetes (1/4%), gestational hypertension (1/4%), prelabor rupture of membranes (1/4%), and thalassemia (1/4%); - Pre-existing diseases not related to pregnancy: Obesity (1/4%) and familial neutropenia (1/4%); - Signs and symptoms (29/100% assessed women): asymptomatic (1/3%); fever at admission (21/72%), cough (18/62%), post-partum fever (6/21%), myalgia (6/21%), dyspnea (5/17%), shivers (4/14%), sore throat (3/10%), chest pain (3/10%), fatigue (2/7%), malaise (1/3%), loss of taste and/or smell (1/3%); - Imaging (25/86% assessed women): suggestive chest CAT scan (24/96%); - Laboratory exams (26/90% assessed women): increased reactive C protein (18/69%), lymphocytopenia (15/58%), leukocytosis (6/23%), neutrophilia (5/19%), interleukin 6 (2/8%), elevated alanine transaminase (1/4%), immunoglobulin G (1/4%) and immunoglobulin M (1/4%); - Delivery (27/100 deliveries): C-section (21/78%)—due to comorbidities associated with pregnancy (10/50%) and (10/50%) due to the infection; vaginal (6/22%). - Maternal death (1/4%).	- 27 newborns - 24 assessed: normal APGAR index (22/92%), normal weight (18/75%), premature (10/41%); - 26 newborns tested: negative for SARS-CoV-2 (24/92%); positive (2/8%); - Neonatal death (2/7%); - Isolation of mother and newborn (13/48%); - Analysis of the placenta of 6 newborns: no alterations infection-related (6/100%).
[34,35,36,37,38,39,40,41,42,43,44,45,46,47,48,49,50,51,52]	Cross-sectional Descriptive	- Settings: China (16/85%), United States of America (1/5%), Italy (1/5%), and Iran (1/5%); - Pregnant women: 546 COVID-19 infected pregnant women, laboratorial diagnosis (409/75%), clinical diagnosis (109/20%), first trimester of pregnancy (39/7%), second trimester of pregnancy (44/8%), third trimester of pregnancy (463/85%), mild or moderate (482/88%) and severe (41/7%) pneumonia; - 429 newborns tested for SARS-CoV-2 (345/81%).	- Pregnancy related comorbidities (398/73% assessed): no comorbidity (139/35%), fetal distress (30/8%), gestational diabetes (30/7%), prelabor rupture of membranes (16/4%), gestational hypertension (12/3%), preeclampsia (7/2%), anemia (5/1%), uterine scar (4/1%), umbilical cord prolapse (4/1%), complete placenta previa (1/0,3%), thalassemia (1/0.3%), and multiple organ dysfunction syndrome/stillbirth (1/0.3%); - Pre-existing diseases not related to pregnancy: hepatitis B infection (4/1%), blood coagulation disorder (2/0.5%), influenza (2/0.5%), hypothyroidism (2/0.5%), schistosomiasis infection (1/0.3%), and hypoproteinemia (1/0.3%); - Signs and symptoms (512/94% assessed women): asymptomatic (55/11%), fever at admission (290/57%), cough (230/45%), dyspnea (65/13%), fatigue (55/11%), myalgia (45/9%), chest pain (38/7%), post-partum fever (23/5%), diarrhea (27/5%), sore throat (19/4%), malaise (5/1%), coryza (2/0.3%), and expectoration (2/0.3%); - Imaging (404/74% assessed women): suggestive chest CAT scan (377/93%); - Laboratory exams (402/74% assessed women): increased Reactive C protein (226/56%), lymphocytopenia (160/40%), leukocytosis (107/27%), elevated alanine transaminase (38/9%), elevated aspartate transaminase (38/9%), neutrophilia (9/2%), immunoglobulin G (5/1%), immunoglobulin M (4/1%), and interleukin 6 (4/1%); - Deliveries (421 deliveries): C-section (273/64%)—due to comorbidities associated with pregnancy (128/46%), due to the infection (74/27%), no information on motive (71/25%); vaginal (148/35%). - Maternal death (7/1%).	- Spontaneous abortion on the 5th week of pregnancy (1) - 429 newborns: normal APGAR index (417/97%), premature (74/17%); - Weight (307/72%): normal birth weight (292/95%), low birth weight (16/5%); - Newborns tested (345/80%): SARS-CoV-2 negative (338/98%), positive (7/2%); - Neonatal death: (8/2%); - 107 pregnant women assessed concerning isolation of which 97 were isolated from newborns; - Analysis of the placenta of 32 newborns: no alterations (32/100%).
[53,54,55,56,57,58,59]	Cross-sectional analytical	- Setting: China (6/86%) and United States of America (1/14%); - Pregnant women: 133 COVID-19 infected pregnant women, laboratorial diagnosis (104/78%), clinical diagnosis (29/22%), first trimester of pregnancy (3/2%), second trimester of pregnancy (23/17%), third trimester of pregnancy (107/80%), mild or moderate pneumonia (132/99%), or severe pneumonia (1/1%); - 108 newborns tested for SARS-CoV-2 (102/94%).	- Pregnancy related comorbidities (116/87% assessed women): no comorbidity (75/65%), gestational diabetes (8/7%), prelabor rupture of membranes (6/5%), gestational hypertension (4/3%), threat of abortion (3/3%), fetal distress (3/3%), uterine scar (2/2%), B-Lynch suture or other compression suture (2/2%), preeclampsia (1/1%), asphyxia (1/1%), and gestational cholestasis (1/1%); - Pre-existing diseases not related to pregnancy: asthma (2/2%) and hepatitis B infection (1/1%). - Signs and symptoms (101/75% assessed women): asymptomatic (18/19%), fever at admission (31/31%), post-partum fever (29/29%), cough (27/27%), dyspnea (5/5%), fatigue (5/5%), chest pain (1/1%); - Imaging (101/75% assessed women): suggestive chest CAT scan (96/95%); - Laboratory exams (74/63% assessed women): increased Reactive C protein (58/78%), lymphocytopenia (57/77%), neutrophilia (57/77%), and leukocytosis (17/23%); - Deliveries (106 deliveries): C-section (59/65%)—due to comorbidities associated with pregnancy (10/17%), due to infection (19/32%), no information on motive (30/51%); vaginal (32/35%).	- 108 newborns: normal APGAR index (107/100%); - Weight (71/76%): normal birth weight (65/92%) and low birth weight (6/8%); - Prematurity: premature (8/7%); - Newborns tested (102/94%): SARS-CoV-2 negative test (102/100%); - Neonatal death (0); - 23 pregnant women were assessed concerning isolation and 20 were isolated from newborns; - Analysis of the placenta of 16 newborns: no alterations (16/100%).
[60]	Case-control	- Setting: China; - Pregnant women: 16 pregnant women infected by COVID-19, laboratory diagnosis (16/100%), in the third trimester of pregnancy (16/100%), mild or moderate pneumonia (16/100%); - 17 newborns: tested for SARS-CoV-2 (3/18%).	- Pregnancy related comorbidities (16/100% pregnant women assessed): no comorbidities (5/31%), gestational diabetes (3/19%), gestational hypertension (2/13%), preeclampsia (1/6%), prelabor rupture of membranes (1/6%), Hepatitis B (1/6%); - Medical background: hypothyroidism (2/13%) and sinus tachycardia (1/6%); - Signs and symptoms (16/100% pregnant women studied): asymptomatic (4/25%), fever at admission (4/25%), post-partum fever (8/50%); - Imaging (16/100% pregnant women studied): suggestive chest CAT scan (10/63%); - Laboratory exams (16/100% pregnant women studied): increased reactive C protein (16/100%), neutrophilia (16/100%), lymphocytopenia (2/13%); - Deliveries (16 deliveries): C-section (14/87%)—does not mention the indication; vaginal (3/13%).	- 17 newborns: normal APGAR index (17/100%), normal weight (14/82%), premature (3/18%); - Newborns tested (3/18%): SARS-CoV-2 negative (3/100%); - Neonatal deaths (0); - pregnant women assessed for isolation (0); - Analysis of the placenta (0).
[61]	Cohort	- Setting: China - Pregnant women: 31 pregnant women infected by COVID-19, laboratory diagnosis (31/100%), in the first trimester of pregnancy (4/13%), in the second trimester (5/16%), in the third trimester (22/71%), mild or moderate pneumonia (21/68%), severe pneumonia (10/32%); - 17 newborns tested for SARS-CoV-2 (17/100%).	- Pregnancy related comorbidities (31/100% pregnant women assessed): no comorbidities (28/90%) and gestational hypertension (1/3%). - Medical background: cardiovascular disease (1/3%) and diabetes (1/3%); - Signs and symptoms (31/100% pregnant women studied): asymptomatic (5/16%), fever at admission (17/55%), cough (15/48%), dyspnea (8/26%), fatigue (6/19%), expectoration (5/16%), myalgia (3/10), diarrhea (2/6%); - Imaging (31/100% pregnant women studied): Suggestive CT scan (31/100%); - Laboratory exams (31/100% pregnant women studied): neutrophilia (10/32%), increased aspartate transaminase (7/23%), - Deliveries (17): C-section (13/76%)—no description on the indication; vaginal (4/24%).	- 17 newborns: normal APGAR index (16/94%), normal weight (16/94%), premature (5/29%); - Newborns tested (17/100%): SARS-CoV-2 negative (17/100%); - neonatal death (0); - Pregnant women analyzed for isolation (0); - Analysis of the placenta (0).

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
