# Peer review of "Effects of COVID-19 Infection during Pregnancy and Neonatal Prognosis: What Is the Evidence?"

_ijerph, 2020, doi:10.3390/ijerph17114176_

Round 1
Reviewer 1 Report
This is a very important manuscript, which can help to assistance to pregnant women and newborns. There is a clear limitation in the study, the fact that the level of evidence is still low, since few studies have been carried out and the majority with reduced quantity or simple design. However, the continuous appearance, especially in the television media, of cases of pregnant women and newborns born by COVID-19, makes us reflect on what measures to adopt.
I have only three concerns about the manuscript:
1. The summary is simple and small. More information can be given on the method and results, the Journal accepts up to 200 words.
2. Materials and methods. This section is very well outlined, with the necessary main guidelines (PRISMA and GRADE) followed, and the appropriate theoretical framework. However, I suggest changing the research question from PICO to PECO strategy (P, target population = pregnant; E, exposure = SARS-Cov2; C, comparison = does not apply; and O, outcomes = Transmission to newborn / fetus ).
3. In the study by Zeng et al., No IgM was found? only IgG?
Congratulations for the study
Author Response
Dear Reviewer, we really appreciate your feedback.
1 The summary is simple and small. More information can be given on the method and results, the Journal accepts up to 200 words
We increased the summary, in order to make it more informative, according to your suggestion.
2. Materials and methods. This section is very well outlined, with the necessary main guidelines (PRISMA and GRADE) followed, and the appropriate theoretical framework. However, I suggest changing the research question from PICO to PECO strategy (P, target population = pregnant; E, exposure = SARS-Cov2; C, comparison = does not apply; and O, outcomes = Transmission to newborn / fetus ).
Thank you very much for that comment. We have corrected the research strategy, according to your indication.
3. In the study by Zeng et al., No IgM was found? only IgG?We increased the summary, in order to make it more informative, according to your suggestion.
Thank you very much for that comment. Both IgG and IgM are studied, but given the mother's transmission behavior by fetus, we chose to discuss only the IgG results.
Reviewer 2 Report
- The aim of this paper was to review current available literature and outline the effects of COVID-19 on pregnancy and neonatal outcomes. The authors reviewed 12 available papers published in 2020--mainly from China--and delineated the clinical presentations, management, and pregnancy and neonatal outcomes. Due to the limited available data to date, recommendations by authors were for cautious and vigilant care of the pregnant patient and their fetus/neonate.
- The strengths of this paper are the ability to compare all available literature to date. Additionally, they had two independent reviewers and a third as an arbitrator.
Limitations include not outlining what the exclusion criteria for including/excluding studies for review.
- Line 114-115: If possible, expand on reasons for cesarean deliveries. Also, specify fetal or neonatal death.
- Line 121: The most common signs and symptoms were fever at both admission and postpartum...
- Line 122:change TC to CT
- Line 147-149: "See for instance" sentence needs to be reworded...maybe author means to compare other viral infections to COVID-19?
- Line 150-151: unclear sentence
- Line 167: cross out very
- Line 168: persisted after childbirth
- Line 170: presented with severe pneumonia. The first woman was in the 35th ...
- Line 171: presented with fever
- Line 172: ICU and developed Multiple Organ...
- Line 173: second women was in the 35th...
- Line 174-175: admission. She reported decreased fetal movements so an urgent C-section...
- 176-178: ...was useful to perform at initial evaluation for admission. Of the pregnant women that received CT scans, 73 had results suggestive of infection.
- Line 184-185: Even so, a single negative result detected via RT-PCR does not preclude a COVID-19 diagnosis, as there are various factors...
- Line 189: change disagreement to discordance
- Line 191: In terms of mode of delivery, most women..
- Line 192: change normal to vaginal
- Line 191-195: was the reason for cesarean delivery due to fetal indications, maternal indications, solely due to respiratory status? provider preference? why were 4 women allowed to have induction of labor? would recommend to relook at data since COVID is not a reason for cesarean...
- Line 204: description of mode of delivery
- Line 210-212: add reference to placental studies
- Line 214: needs reference
- Line 216-217: recommendations are that C-sections be performed in a negative pressure room...
- Line 215-218: add reference
- Line 218-220: add reference
- Line 229: Consider changing Nonetheless to In summary,
- Line 231: in terms of mode of delivery
Author Response
Dear reviewer, thank you very much for the comments and corrections throughout the text. We carefully looked at each of the grammatical or formatting errors pointed out, reviewing them and highlighting the changes, as well as adding references where the reviewer pointed out that it was necessary. Regarding the other suggestions or doubts, we clarify that:
1. Limitations include not outlining what the exclusion criteria for including/excluding studies for review.
-Thanks for this comment, we try to make it as clear as possible what the inclusion criteria are (Be one of the chosen databases, be an observational epidemiological study or case reports addressing the clinical conditions of mother-RN/fetus pairs and including primary data of patients over 18 years old) and exclusion (Manuscripts that brought only data from pregnant women, or only RN, or that did not address the period of delivery).
2. Line 114-115: If possible, expand on reasons for cesarean deliveries. Also, specify fetal or neonatal death.
-We accurately detail this whenever possible. However, we depend on what the authors put in, the results and discussion section provides more details on that.
3. Line 191-195: was the reason for cesarean delivery due to fetal indications, maternal indications, solely due to respiratory status? provider preference? why were 4 women allowed to have induction of labor? would recommend to relook at data since COVID is not a reason for cesarean...
-The reasons for cesarean section seem to be related to the mother's severity, the team's fear and the local protocol. However, we discuss all these aspects in depth in the text.
Thank you.
Reviewer 3 Report
The authors are to be congratulated for the effort to put together the data published in small case series and observational studies.
Several issues however arise.
One of the biggest problem in dealing with pregnant women in the light of the COVID-19 pandemic is that we almost know nothing about the disease. So, with the exact same aim of achieving the optimal pregnancy outcome for all mothers and infants, one can decide to rather err on the cautious side (with lots of activism, performing unproven therapies, causing iatrogenic harm by unnecessary termination of pregnancy, preterm induction of labor, separations of mothers and infants, avoiding breast feeding and and and). Or one can decide to err on the "risky" side not doing nothing of the things mentioned above and treating mothers and infants as if they had another viral illness - with the risk of a serious course of disease. Which approach turns out to really be the more dangerous one is unclear, because we do not have the data yet. It is pretty clear however, that there are several risks in the context of COVID-19. That risks brought by the disease itself is only one part of these risks. Another one is lack of general resources because all resources are relocated to care for COVID-19 patients; poorer surveillance of pregnancy because of restricitions (by medical staff or by pregnant women themselves) to come to appointments within the hospital, by reduced number of checks and and and. And third by iatrogenic harm; induced preterm delivery
Did the authors check for double reporting of identical patients in different publications?
A substantial numbers of case reports, case series, cross-sectional studies and also other systematic reviews is not reported (many of them published after april 1st).
Several reviews have been published recently (Mullins, Schwartz, Peyronnet, Placais, Zaigham, Della Gatta, DiMascio, Karimi-Zarchi, Parazzini, Yan J (AJOG)). Elshafeey (Int J Gynaecol Obstet) covers much more studies, including studies fomr netherlands (nvog.nl), italy (Ferrazzi) and the US (Breslin, Juusela) amongst many other case reports. Please clarify the added value of this study. Studies neither included in this report nor in Elshafeey include these of Browne PC, Buonsenso D, Cao D, Chen L (NEJM), Khan S, Li L (AJR), Lu D (J Med Virol), Sutton D, Peng Z (J Inf Public Health), Sharma KA, Tekbali A, Vlachodimitropoulou Koumoutsea E, Xia H (Br J Anaesthesiol), Yang H (J Infect), Yang P (J Clin Virol), Yu N (Lancet Inf Dis), Zamaniyan M.
The conclusions do not match the results. Where is the data indicating that pregnant women must be cautious and vigilant (other that any pregnant woman should be in any other situation)? What are the important implications that infected women can face, especially in terms of the type of childbirth? It seems, the highest risk for pregnant women might not be the virus itself but that they might meet doctors that have read and follow recommendations which are based on fears and not on data. The risk for iatrogenic preterm birth, unnecessary c-sections, CT-scans without a consequence, off label drug administration with potential harm, no proven benefit and even without the necessity seems to be substantial hazards, possibly more dangerous than the disease that shall be treated.
Something went wrong with the references between table and text. Lee (16) should be Ref No 24. Liu (17) is probably Ref No 16
Minor comments:
Introduction can be shortened: reiteration of droplet transmission and survival on surfaces is not relevant in the context of this article (and there is more recent literature about surface viability), explanation of disease severity is most likely incomplete at least (also here is more recent data available).
Please have language checked by native english speaker.
Please check Ref. 1 and give original Reference to WHO statement.
Page 2 line 48: Please give evidence for increased risk of morbidity and mortality for viral diseases.
Page 2 line 51: Please give more robust evidence that fever and tachycardia in general are less tolerated and lead to complications by itself in pregnant women.
Page 2 line 64: From the objective I assume that outcome of interest is not only fetal but also maternal outcome?
Page 2 line 80: This paragraph is most likely incomplete, as diagnostic data was gathered also. I would suggest to shorten it. Did you look into loss of smell or taste?
Page 3 line 98: Four case reports, but only three references (13, 14, 15) given.
Page3 line 98: Please state how many women are described in these four case reports. One woman-infant dyad in each publication?
Page 3 line 106: 40 women represent 33.6% of which sample?
Page 3 line 107: 60% of the women described in studies 16-20 had at least one comorbodity?
Page 3 line 113: 28 women represent 93.3% of which sample?
Page 3 line 118: 72 women represent 60.5% of which sample?
Page 3 line 120: The 45 other women had no comorbodities or multiple comorbidities?
Page 3 line 122: Chest CT
Page 4 line 123 and 124: Brackets are missing for ref 23
Page ? line 145: The whole fertile period is broader than 22-42yrs.
line 147: Is it reasonable to believe that there is a high risk of embryotoxicity due coronaviridae? (what is known about first trimester infection with SARS and MERS and other Coronavirus-Infections?)
Line 153: Please give reference for the statement, that respiratory disease (viral or other?) is associated with high rate of c-section, prematurity, low APGAR score and low birth weight.
Line 155: Please quantify what is meant by "high risk of severe pneumonia during this period". And it is "high" as compared to which group? To pregnant women in the first or second trimester? To nonpregnant women?
Line 158: Ref 28 does not support the statement of high risk for pregnant women. It does not even contain the word "pregnant".
Line 169: This information is misleading. Most women in the data published here were diagnosed as having mild pneumonia. We don?t know nothing about those who did not appear in these observative studies. Within theses studies, tests were mostly performed in women with symptoms, maybe severe enough to present in hospitals. It is possible, that a significant proportion of pregnant women did not even develop symptoms or only mild symptoms for which they did not seek assistance.
Line 192: What is meant by "recent literature tends to lean toward c-sections?" In general? Please give reference. Also for COVID-19-patients, we need to differentiate, whether this advice is based on expert opinion or on data. The fact that c-section is performed tells nothing about whether it was necessary. At least for mild or moderate disease, there is no evidence to believe, that c-section will improve the course of the disease. Antiviral drug administration does not indicate the need for termination of pregnancy, especially, if there is no antiviral treatment shown to be effective for the mother. Also antibiotics don't need to be withheld just because of the pregnancy. If indicated, they can be given nonetheless. Antibiotics without bacterial superinfection are not likely to help.
Line 206: SARS-CoV-2
Line 216: These unsubstantiated recommendations should not be repeated. The scope of this review was to scan and assess the evidence about the importance of COVID-19 infections in pregnant women - not to make recommendations in the absence of data. There is more than enough of these recommendations.
Line 223: Which of 21-23 are the two?
Author Response
Dear reviewer, thank you for your extensive and in-depth review! Below we point out the changes made and the justifications for the situations in which they could not be made.
1. Did the authors check for double reporting of identical patients in different publications?
-Thanks for this comment, however this is one of the main limitations of our findings. It is almost impossible in a review study without identifying patients to realize and claim that someone has been included twice. We put this as a limitation in the paper.
2. A substantial numbers of case reports, case series, cross-sectional studies and also other systematic reviews is not reported (many of them published after april 1st). Several reviews have been published recently (Mullins, Schwartz, Peyronnet, Placais, Zaigham, Della Gatta, DiMascio, Karimi-Zarchi, Parazzini, Yan J (AJOG)). Elshafeey (Int J Gynaecol Obstet) covers much more studies, including studies fomr netherlands (nvog.nl), italy (Ferrazzi) and the US (Breslin, Juusela) amongst many other case reports. Please clarify the added value of this study. Studies neither included in this report nor in Elshafeey include these of Browne PC, Buonsenso D, Cao D, Chen L (NEJM), Khan S, Li L (AJR), Lu D (J Med Virol), Sutton D, Peng Z (J Inf Public Health), Sharma KA, Tekbali A, Vlachodimitropoulou Koumoutsea E, Xia H (Br J Anaesthesiol), Yang H (J Infect), Yang P (J Clin Virol), Yu N (Lancet Inf Dis), Zamaniyan M.
-Following our inclusion and exclusion criteria (which were very careful to provide valid and robust evidence) we updated the material, adding 32 more texts from a series of countries (China, [13-15,17,20-23 , 25,28,31-41,43,45-54] 44 USA, [19,42] 42 Italy, [44], and 1 and 1 pregnant woman in Asia, [16] Honduras, [18] Australia, [ 24] Turkey, [26] Iran, [27] Spain [29] and Peru. [30])
3. The conclusions do not match the results. Where is the data indicating that pregnant women must be cautious and vigilant (other that any pregnant woman should be in any other situation)? What are the important implications that infected women can face, especially in terms of the type of childbirth? It seems, the highest risk for pregnant women might not be the virus itself but that they might meet doctors that have read and follow recommendations which are based on fears and not on data. The risk for iatrogenic preterm birth, unnecessary c-sections, CT-scans without a consequence, off label drug administration with potential harm, no proven benefit and even without the necessity seems to be substantial hazards, possibly more dangerous than the disease that shall be treated.
-Dear, this is something complicated to address. In fact, our study shows that there are many papers but little evidence, that there is contamination but doubts about the possibility or route of transmission, reinforcing even that in conditions where the mother was not tested in time, she was allowed to breastfeed the baby and so contaminate it. Thus, we believe that our greatest contribution is to request attention, especially from health professionals to mothers and babies since the beginning of pregnancy. Let us know if this is still not the best approach to your point of view and if we can improve.
Something went wrong with the references between table and text. Lee (16) should be Ref No 24. Liu (17) is probably Ref No 16
-Its was corrected.
Minor comments:
Introduction can be shortened: reiteration of droplet transmission and survival on surfaces is not relevant in the context of this article (and there is more recent literature about surface viability), explanation of disease severity is most likely incomplete at least (also here is more recent data available).
--Its was corrected.
Please have language checked by native english speaker.
-Its was revised.
Please check Ref. 1 and give original Reference to WHO statement.
-Its was corrected.
Page 2 line 48: Please give evidence for increased risk of morbidity and mortality for viral diseases.
-Its was corrected.
Page 2 line 51: Please give more robust evidence that fever and tachycardia in general are less tolerated and lead to complications by itself in pregnant women.
Page 2 line 64: From the objective I assume that outcome of interest is not only fetal but also maternal outcome?=
-Yes, its was corrected.
Page 2 line 80: This paragraph is most likely incomplete, as diagnostic data was gathered also. I would suggest to shorten it. Did you look into loss of smell or taste?
-Yes, its was corrected. All sections with misspellings were redone.
line 147: Is it reasonable to believe that there is a high risk of embryotoxicity due coronaviridae? (what is known about first trimester infection with SARS and MERS and other Coronavirus-Infections?)
-Thanks for that question! Please see our discussion of this in the paper: Regarding the pregnant women's age, the fertile period, and the length of pregnancy, the studies analyzed showed a wide variation and a lack of evidence of infection by SARS-CoV-2 during the first and second trimester of pregnancy. It can be inferred that, according to the low prevalence of severe infection among pregnant women (45/10%), many of them could be asymptomatic and/or with mild symptoms, without the need for hospital care. Corroborating this data, a Norwegian cohort study of 1258 pregnant women during the influenza pandemic in 2009 showed that 226 (18%) had influenza (H1N1) and only three were hospitalized. It is noteworthy that most pregnant women were in the first or second trimester of pregnancy. The study authors state that there is little evidence that mild influenza during pregnancy is associated with an increased risk of preeclampsia as well as premature and small birth-weight babies for gestational age [55].
Regarding the Middle East Respiratory Syndrome (MERS), there are limited data on the prevalence and clinical characteristics of MERS during pregnancy, birth, and the postnatal period. A systematic review with meta-analysis recovered seven studies, which did not report spontaneous abortion. The rate of premature birth was 32.1% (3 of 11), all occurring before 34 weeks of gestation. Preeclampsia was described in 19.1% (1 of 7), however, no cases of premature rupture of membranes or restricted fetal growth were reported. The rates of cesarean delivery and perinatal death were 61.8% (5 of 8) and 33.2% (3 of 10), including two stillborn and one neonatal death (4 hours after the birth of an extremely premature baby), respectively. There were no reports of fetal distress, Apgar score <7 at 5 minutes, neonatal asphyxia, and admission to the neonatal intensive care unit (ICU). Finally, signs of vertical transmission were not found during the follow-up period in any of the newborns [56].
According to the above, it appears that the limited data on infection with the new coronavirus in early pregnancy may be related to the absence of tests performed during this period, as Apgar there are asymptomatic pregnant women, with mild signs and symptoms and there is no screening of these pregnant women since the beginning of pregnancy, this infection may go unnoticed or be detected only after delivery. Thus, we suggest that tests for COVID-19 should be performed as a routine in prenatal care.
All other suggestions have been reviewed and are in the discussion. Due to the limited character and in order to optimize the reading of the opinion, we invite the reviewer to observe it directly in the paper.
Thank you again.
Round 2
Reviewer 3 Report
The manuscript is significantly expanded now. However, it is still not up to date with regard to the date of submission of the revised version. Important pieces of literature are still missing, like the description of a probable vertical transmission by Maksim Kirtsman et al (doi 10.1503/cmaj.200821).
The manuscript needs more structure to be readable after the significant expansion. As of now, the manuscript reads like many small pieces put together without an integrating synthesis.
The language deteriorated to a degree that makes it difficult to grab the message.
The conclusion (as well in the abstract as in the manuscript) needs to be rewritten. What is the bottom line that evidence tells us? What is (as far as we know) the most likely scenario for women being hit by COVID-19 in the pregnancy? If most likely benign: Can serious courses be ruled out? If no: are there predictors for a serious course? Do we know anything about the most likely modes of transmission to the infant? vertical in utero? after birth from the mother? From other caregivers in the hospital? from other household members? can we estimate the degree of iatrogenic excess morbidity and mortality and compare it to the burden of the disease itself? Do we know how many effort should be made to reduce infections of the newborn? Is separation justified and in which cases? Are there factors that predispose for a serious course in the newborn infant?
If the manuscript wants to be more than a (out of date) collection of studies it needs to try to answer at least some of these questions in the discussion - and it should clearly distinguish between available evidence and interpretation by the authors.
Once the manuscript is accepted or re-submitted the authors should include very recent information about virus found in breast milk (doi.org/10.1016/S01406736(20)311818).
Author Response
- The manuscript is significantly expanded now. However, it is still not up to date with regard to the date of submission of the revised version. Important pieces of literature are still missing, like the description of a probable vertical transmission by Maksim Kirtsman et al (doi 10.1503/cmaj.200821).
- Dear reviewer, thank you very much! We wanted to ask for your understanding regarding what we can consider “updated”. There is a large production on SARS-Cov-2 in the world, so we will not be able to exhaust in real time all the manuscripts that come out, especially on a topic that publishes around 1000 texts daily according to a recent study.
- A review should exhaust the knowledge within the proposed time frame, which was in ours until May 5th. The text to which the reviewer referred was published on the 14th just one day before we submitted the final and significantly expanded version of the review to IJERPH. Therefore, it would be impossible to read, retain the knowledge, remove the necessary information, put it in the text, share it with all the authors and wait for their feedback. Thus, we ask for the understanding to read our results within such logic. Thank you.
- The manuscript needs more structure to be readable after the significant expansion. As of now, the manuscript reads like many small pieces put together without an integrating synthesis.
-Thanks for that comment. Our manuscript is "hostage" to the great diversity of approach and knowledge published worldwide. Thus, “exhausting knowledge” did not lead us to structure the manuscript in the best possible way, being that we found that we had to present 50 manuscripts. However, we try to heed your suggestions, putting subtopics and significantly wiping out results and discussion.
- The language deteriorated to a degree that makes it difficult to grab the message.
-We apologize. The manuscript was analyzed by a native professional in the new version.
- The conclusion (as well in the abstract as in the manuscript) needs to be rewritten.
-The two conclusions have been rewritten.
4.1 What is the bottom line that evidence tells us?
-The final result is inconclusive compared to the current scenario, mainly due to low evidence. This is willing:
- Lines 25-29: At the conclusion of the summary: “We cannot rule out potential worsening of the clinical condition of pregnant women infected with SARS-CoV-2, whether it is associated with comorbidities or not, due to the occurrence of respiratory disorders, cardiac rhythm disturbances, and acid-base imbalance, among others. We recommend relentless monitoring of all pregnant women in addition to testing them before delivery or the first contact with newborns”.
- Lines 337-347 (In Discussion): this review has strengths such as presenting a comparison of all studies available so far, compiling scattered data and grouping them more clearly for making analyses and inferences and, based on what was assessed, reporting that there is no evidence of vertical transmission thus far, as there are knowledge gaps concerning the care taken during and after delivery, and a lack of suitable biological samples for testing the SARS-CoV-2.
- In their conclusions (Line 348 to 362): The results show that the potential worsening of the women infected with SARS-CoV-2 cannot be ruled out, whether the infection is associated with comorbidities or not. As documented here, there is a risk of women developing respiratory disorders, having cardiac rhythm disturbances, or acid-base imbalance, among others. The risk of postpartum hemorrhage and premature delivery is significant, which is why C-sections were widely used. Relentless monitoring is recommended for pregnant women who report signs and symptoms suggested of COVID-19, especially at a time when women in countries with radical isolation measures have difficulty in attending regular prenatal care. Those who cannot be tested before delivery should avoid having contact with the baby and wear masks to decrease spread of the disease. Regarding the risk for babies, there is no concrete evidence of vertical transmission, though one cannot discard this possibility. Nonetheless, cases of respiratory diseases, abnormal Apgar indexes, and mild pneumonia were reported. Fortunately, all the babies who tested positive for the infection recovered fully and rapidly. A rigorous assessment of the newborns’ clinical signs is recommended, as well as chest CT scans within three days after birth.
4.2: What is (as far as we know) the most likely scenario for women being hit by COVID-19 in the pregnancy? If most likely benign: Can serious courses be ruled out? If no: are there predictors for a serious course?
-These questions were answered:
- In the Abstract (Lines 24 to 29): We cannot rule out potential worsening of the clinical condition of pregnant women infected with SARS-CoV-2, whether it is associated with comorbidities or not, due to the occurrence of respiratory disorders, cardiac rhythm disturbances, and acid-base imbalance, among others. We recommend relentless monitoring of all pregnant women in addition to testing them before delivery or the first contact with newborns.
- In discussion (Lines 226 to 268: .... The presence of comorbidities related to pregnancy does not seem to directly influence the adverse outcomes of pregnant women and their newborns, as the two neonatal deaths were of mothers without comorbidities, but who, for some reason developed severe pneumonia. However, it is observed that gestational diabetes and fetal distress were the most prevalent comorbidities, showing that the conditions of the fetus should be carefully evaluated, especially, in those asymptomatic and without comorbidities. Thus, the absence of comorbidities may directly influence the care provided and attention paid by professionals to pregnant women, exposing some of them to a worse prognosis.[65]
- In conclusion (Lines 349 to 357): The results show that the potential worsening of the women infected with SARS-CoV-2 cannot be ruled out, whether the infection is associated with comorbidities or not. As documented here, there is a risk of women developing respiratory disorders, having cardiac rhythm disturbances, or acid-base imbalance, among others. The risk of postpartum hemorrhage and premature delivery is significant, which is why C-sections were widely used. Relentless monitoring is recommended for pregnant women who report signs and symptoms suggested of COVID-19, especially at a time when women in countries with radical isolation measures have difficulty in attending regular prenatal care. Those who cannot be tested before delivery should avoid having contact with the baby and wear masks to decrease spread of the disease.
4.3: Do we know anything about the most likely modes of transmission to the infant? vertical in utero? after birth from the mother? From other caregivers in the hospital? from other household members? Do we know how many effort should be made to reduce infections of the newborn? Is separation justified and in which cases? Are there factors that predispose for a serious course in the newborn infant?
- The studies are quite divergent in this sense, and none of them is able to state with 100% accuracy that there was transmission, since a series of limitations is raised with respect to: isolation of the baby's mother, use of protective equipment by the mother and staff, use of negative pressure, breastfeeding ... among others. However, we seek to discuss all possible transmissions and the modes / routes involved in discussion: 295 – 331.
Once the manuscript is accepted or re-submitted the authors should include very recent information about virus found in breast milk (doi.org/10.1016/S01406736(20)311818).
-Thanks, although the text does not meet the inclusion criteria, we added your important findings to the results!
Thanks again for the valuable contributions of the reviewer. We hope that the current version can be accepted!